# Nitric Oxide Overproduction by *cue1* Mutants Differs on Developmental Stages and Growth Conditions

**DOI:** 10.3390/plants9111484

**Published:** 2020-11-04

**Authors:** Tamara Lechón, Luis Sanz, Inmaculada Sánchez-Vicente, Oscar Lorenzo

**Affiliations:** Department of Botany and Plant Physiology, Instituto Hispano-Luso de Investigaciones Agrarias (CIALE), Facultad de Biología, Universidad de Salamanca, C/Río Duero 12, 37185 Salamanca, Spain; tlg@usal.es (T.L.); lusan@usal.es (L.S.); elfik@usal.es (I.S.-V.)

**Keywords:** nitric oxide homeostasis, *cue1/nox1*, reactive nitrogen species, germination, root development, stress responses, sugar metabolism

## Abstract

The *cue1* nitric oxide (NO) overproducer mutants are impaired in a plastid phosphoenolpyruvate/phosphate translocator, mainly expressed in Arabidopsis thaliana roots. *cue1* mutants present an increased content of arginine, a precursor of NO in oxidative synthesis processes. However, the pathways of plant NO biosynthesis and signaling have not yet been fully characterized, and the role of CUE1 in these processes is not clear. Here, in an attempt to advance our knowledge regarding NO homeostasis, we performed a deep characterization of the NO production of four different *cue1* alleles (*cue1-1*, *cue1-5*, *cue1-6* and *nox1*) during seed germination, primary root elongation, and salt stress resistance. Furthermore, we analyzed the production of NO in different carbon sources to improve our understanding of the interplay between carbon metabolism and NO homeostasis. After in vivo NO imaging and spectrofluorometric quantification of the endogenous NO levels of *cue1* mutants, we demonstrate that CUE1 does not directly contribute to the rapid NO synthesis during seed imbibition. Although *cue1* mutants do not overproduce NO during germination and early plant development, they are able to accumulate NO after the seedling is completely established. Thus, CUE1 regulates NO homeostasis during post-germinative growth to modulate root development in response to carbon metabolism, as different sugars modify root elongation and meristem organization in *cue1* mutants. Therefore, *cue1* mutants are a useful tool to study the physiological effects of NO in post-germinative growth.

## 1. Introduction

Since the establishment of nitric oxide (NO) as an endogenous signaling molecule in plants over twenty years ago [1,2,3,4], a lot of progress has been made towards understanding NO synthesis and signaling in these organisms. In contrast to other eukaryotes, at least seven different sources of NO generation have been characterized in plants [5,6,7,8]. While in mammalian cells NO is synthesized from either nitrite or arginine oxidation in a reaction catalyzed by the enzyme NO synthase [9]; in plants, the existence of this last pathway is controversial [5,10]. Plant NO synthesis is controlled by a number of enzymatic synthesis reactions, catalyzed by the enzyme nitrate reductase, by the mitochondrial electron transport chain, or by the enzyme xanthine amine oxidoreductase during anaerobic conditions (reviewed in [5,6,7,8,9,10,11]). Moreover, NO can also be produced non-enzymatically, from acid solutions of nitrite in the presence of compounds that can act as antioxidants [12]. However, we only identified some of the components implicated in the different pathways and it is still unknown how they interact, how they fit in the larger context of carbon and nitrogen metabolism, and how much they contribute to the general NO homeostasis of the plant. The concentration of NO is also tightly regulated through the interactions of NO with other compounds, such as reactive oxygen species (ROS), proteins or lipids. Because of its physicochemical characteristics, NO exerts its functions mainly through modification of these molecules, leading to changes in protein activity, gene expression, and modulation of the redox environment, both during physiological and stress responses (reviewed in [6,7,8,13]). In an attempt to answer these fundamental questions about when and where NO is produced, researchers have used both pharmacological and genetic approaches.

The use of pharmaceutical NO donors such as sodium nitroprusside (SNP), *S*-nitroso-*N*-acetyl-d,l-penicillamine (SNAP), or *S*-nitrosoglutathione (GSNO), although very extended, does not always replicate the endogenous effects of NO. SNP is in fact a nitrosonium cation donor that also generates cyanide [14]. On the other hand, SNAP and GSNO, in addition to releasing NO, can act mainly through trans-*S*-nitrosation reactions. Furthermore, the application of NO donors might result in nitrosative stress since there have been very few attempts to understand the kinetics of NO generation *in planta* by NO donors [15,16,17]. Thus, the use of mutants with altered endogenous NO content seems to be a more suitable way of assessing NO-modulated responses.

In *Arabidopsis thaliana*, only four groups of NO overproducer mutants have been described so far, *gsnor1/hot5*, *glb*, *argah*, and *cue1/nox1* mutants. *gsnor1/hot5* mutants accumulate GSNO, since they are defective in the enzyme *S*-nitrosoglutathione reductase 1 (GSNOR1) [18], responsible for the degradation of GSNO, a stable NO reservoir, to glutathione disulfide (GSSG) and ammonium [19]. Thus, *gsnor1* mutants accumulate both NO and GSNO [20]. The *glb* mutants are also impaired in a NO scavenging system, since they have reduced levels of non-symbiotic hemoglobins, which usually eliminate NO by binding it to their heme group [21]. The last two groups, *argah* and *cue1*, are both thought to be implicated in the oxidative synthesis of NO from arginine [22,23]. *argah* mutants have decreased arginase activity, which results in an increase in the available arginine pool, since arginases control the catabolism of this amino acid. This, in turn, leads to the accumulation of NO in these mutants [22].

The first *A. thaliana cue1* mutant was isolated in a screening to identify new mutants in light signaling components with an altered light-regulated expression of nuclear genes [24]. *CUE* stands for ‘chlorophyll *a/b* binding protein (*CAB*) underexpressed’ because mutations at this locus result in expression defects of photosynthesis genes in mesophyll cells, such as the light-harvesting chlorophyll *a/b*-binding protein 1 (*LHCB1*) of photosystem II, formerly known as *CAB*. Besides *cue1*, another eight *cue* mutants were isolated [25]. All of them are defective in greening and present an altered mesophyll structure. These defects result from a delayed differentiation of chloroplasts and a reduction in plastid size and granal stack size, along with defective etioplast development [25]. In *cue1*, *LHCB1* is expressed at low levels in the mesophyll cell layers but at wild-type levels in the bundle sheath cells, which causes a striking reticulate leaf phenotype with pale-green mesophyll cells and dark-green veins. At the same time, it entails a severe deficiency in the establishment of photoautotrophic growth because of the lack of sufficient carotenoids and chlorophylls, especially during early leaf development in response to light [24].

Of the nine *CUE* genes, only *CUE1* does not directly take part in the phytochrome-controlled expression of photosynthetic genes [25,26]. Instead, *CUE1* encodes plastid phosphoenolpyruvate (PEP)/phosphate translocator (PPT) expressed mainly in roots, but also in leaves and flowers [27]. PEP is the precursor for the shikimate pathway of aromatic amino acids and can be utilized as an alternative source for ATP in non-photosynthetic plastids. Most plastids either lack or have a very low expression of the complete set of glycolytic enzymes for the conversion of hexose and triose phosphates into PEP, so glycolysis cannot proceed further than 3-phosphoglycerate. PPT is thus the only source of PEP in the chloroplast stroma [28]. In line with this role, *cue1* mutants present an altered content of several amino acids and secondary metabolites. Among them, there is an increase in nitrate, arginine and two products of arginine catabolism, citrulline and urea [27].

Both nitrate and arginine are precursors of NO biosynthesis [29], so it was to be expected that *cue1* mutants also showed increased NO content. Indeed, in screening for NO overproducer (*nox*) mutants, the locus *NOX1* was identified as *CUE1* [23]. The isolation of putative *nox* mutants was based on their hypersensitivity to root growth inhibition by NO donor SNP. *nox1*/*cue1* was confirmed to have higher levels of NO in rosette leaves when analyzed with NO-sensitive dye 4,5-diaminofluorescein diacetate (DAF-2DA). Since then, *cue1* mutants have been described to have delayed flowering [23], smaller rosettes, leaves and cotyledons [30], increased stomatal development [31], reduced root length and meristem size [32,33,34], decreased mitosis and increased endoreduplication [32,35], reduced auxin response [32], decreased cytokinin content [36], higher iron uptake [37], better copper tolerance [38], and less pathogen resistance [39,40,41].

Because *cue1* mutants display a complex pleiotropic phenotype, understanding the role of *CUE1*/*PPT1* in NO synthesis and signaling is not trivial. Here, we sought to establish the growth conditions in which *cue1* mutants overproduce NO and to identify which phenotypes can be unequivocally ascribed to an altered NO homeostasis in order to close the current gap between pharmacological and genetic studies. For this purpose, we analyzed the phenotype of four different *cue1* alleles during physiological processes and stress responses in which NO is known to have an important role: promotion of germination [42,43,44,45], inhibition of primary root elongation [23,32,33], and improvement of salt stress resistance [46,47,48]. This was done alongside the in vivo NO imaging and quantification of the NO content of each mutant by spectrofluorometry using the fluorescent NO-sensing dye 4-amino-5-methylamino-2′,7′-difluorofluorescein diacetate (DAF-FM DA) according to the method described in [49], to find out if there is a link between NO levels and the observed phenotypes. Furthermore, we analyzed the production of NO in different carbon sources to improve our understanding of the interplay between carbon metabolism and NO homeostasis, for which *cue1* mutants are an excellent model, as the primary role of *PPT1* is likely to be related to glycolysis and gluconeogenesis in the plastids.

## 2. Results

### 2.1. cue1 Mutants Accumulate Wild-Type NO Levels during Early Post-Germinative Plant Development

A rapid increase in NO levels appears in the endosperm of *A. thaliana* seeds after imbibition [50]. NO then promotes germination both by relieving dormancy [42,51] and by directly promoting embryo growth [45]. In order to study the response of *cue1* mutants during germination, we chose four different knockdown and knockout alleles that have been extensively used in the literature and are known to have increased NO in rosette leaves [23,31,36,39,40] (Figure 1A). Three of the alleles come from a mutagenized Col-0 line, *cue1-5*, *cue1-6*, and *nox1-1*. *cue1-5* is a weak allele with an Arg to Cys point mutation [27], *cue1-6* is a strong allele with a premature stop codon instead of a Trp caused by a point mutation [27], and *nox1-1* is a knockout mutant which was obtained by fast neutron mutagenesis and lacks most of the genomic *CUE1*/*PPT1* sequence [23]. *cue1-1* is another deletion mutant, but it was obtained by gamma radiation mutagenesis of a transgenic line, pOCA108 (Be-0), that is also an alcohol dehydrogenase null mutant [24].

Although these mutants are routinely used as constitutive NO-overproducer mutants, there is no data on their NO levels during early plant development. While the link between *PPT1*, light signaling, and NO production is not understood, it is known that CUE1 strongly regulates the expression of *LHBC* and other photosynthetic genes when plants are initially exposed to light, but *cue1* mutants show a greater degree of plasticity at later stages of development [27]. Therefore, we quantified the NO content of the four alleles and their respective wild-type controls 4 days after completion of stratification (das), once the radicle was visible in most of the seeds in the population. Contrary to what was expected, all four *cue1* alleles had similar or less DAF fluorescence than their respective controls, suggesting that the mutants do not overproduce NO at this developmental stage. *cue1-6* and *nox1* had the same fluorescence intensity than Col-0 (0.77 ± 0.01 a.f.u./μg protein and 0.74 ± 0.01 a.f.u./μg protein compared to 0.72 ± 0.05 a.f.u./μg protein), whereas *cue1-5* (0.49 ± 0.05 a.f.u./μg protein) had 36% less than Col-0 (*p* < 0.001) and *cue1-1* (0.68 ± 0.06 a.f.u./μg protein), 25% less than pOCA108 (0.91 ± 0.05 a.f.u./μg protein; *p* < 0.001) (Figure 1B).

High salinity delays germination and impairs seedling establishment [52]. Endogenous NO is increased when plants are exposed to salt stress and might act as an antioxidant by quenching the ROS produced in response to salinity [53,54,55,56]. In order to see whether *cue1* mutants would accumulate NO during salt stress, we indirectly quantified the NO content of 4-day-old seedlings grown on a medium supplemented with 100 mM NaCl through quantification of DAF fluorescence. This salt concentration was chosen because it has been previously shown that it is enough to cause salt stress to *A. thaliana* seedlings [53]. Under our growth conditions, the DAF fluorescence significantly decreased in all the examined lines after salt stress, suggesting an inability to overproduce NO at this stage (Figure 1). In this case, *cue1-5* had 0.4 ± 0.02 a.f.u./μg protein, only 23% less than Col-0, which had 0.52 ± 0.03 a.f.u./μg protein. The rest of the lines had similar fluorescence levels to those of Col-0, ranging from 0.44 ± 0.02 a.f.u./μg protein to 0.47 ± 0.01 a.f.u./μg protein. The greatest decrease in the presence of high salinity was observed in pOCA108, which had 48% of the fluorescence intensity it accumulated in control conditions.

### 2.2. NO Is Necessary to Maintain Germination Vigor

Given the role of NO in the promotion of seed germination [45], we decided to analyze the germination rate of the different *cue1* alleles. While most of the Col-0, *cue1-6*, *nox1-1*, pOCA108, and *cue1-1* seeds had fully germinated after 4 das (between 96.5% and 99.5% of the population), the maximum germination (G_max_) of *cue1-5* was only 67%, in accordance with the decrease in DAF fluorescence and supporting a reduced NO content at this stage (Figure 2A). In the presence of high salinity, the germination of all the lines decreased consistently with a reduction in endogenous NO. This germination delay was especially apparent for *cue1-5*, which had a G_max_ of 48% while the rest of the lines had similar G_max_ that varied between 66% and 88% (Figure 2A). Despite the stark decrease in DAF fluorescence after salt stress (Figure 1), the germination of pOCA108 was almost unaffected by the stress, with only an 11% decrease in its germination rate with respect to control conditions.

The speed and uniformity of germination were analyzed using t_50_ and U_7525_. t_50_ is a parameter that summarizes the time required for 50% of the viable seeds to germinate [57]. We found an inverse correlation between DAF fluorescence and speed germination parameter t_50_ (*r* = −0.94, *p* < 0.001). It took 69.99 ± 4.96 h for half of the *cue1-5* seeds to germinate, whereas the t_50_ of Col-0 was only 49.12 ± 1.69 h, 30% faster than the mutant. Equally, the t_50_ of pOCA108 was 44.92 ± 1.04 h, 19% faster than that of *cue1-1*, 55.19 ± 1.8 h (Figure 2B). U_7525_ measures the time interval between the germination of 25% and 75% of the viable seeds. A lower U_7525_ indicates greater uniformity [58]. The analysis of uniformity supported an inverse correlation between NO levels and germination (*r* = −0.87, *p* < 0.001). The U_7525_ of *cue1-5* was 22.65 ± 2.64 h, 13.16 h longer than its control, while on the other end of the spectrum pOCA108 had a U_7525_ of 6.9 ± 1.03 h (Figure 2B). These results suggest that a certain level of NO is necessary for seeds to germinate uniformly once dormancy is broken. Cumulative germination over 7 das showed a similar exponential curve for Col-0 and pOCA108, with a steeper slope for pOCA108, the line that showed the highest endogenous NO levels. On the other hand, all *cue1* mutants showed a longer lag phase that lasted up to 48 h after completion of stratification, which suggests that endogenous NO levels might be even lower in *cue1* mutants prior to radicle emergence (Figure 2C).

In the presence of high salinity, the t_50_ and U_7525_ of all the lines were similar (Figure 2D), in agreement with their similarity in NO production (Figure 1). While *cue1-1* seemed to have the highest t_50_ at 87.0 ± 6.1 h, this difference was not significant when compared to its control or other lines, which showed t_50_ ranging from 73.4 ± 3.3 h to 82.3 ± 2.0 h. Even though germination was generally slower under salt stress, uniformity of germination was more stable for all genotypes. Control lines had slightly more uniform germination with a U_7525_ of 15.0 ± 1.4 h for Col-0 and 13.8 ± 1.9 h for pOCA108, while the U_7525_ of the *cue1* mutants ranged between 15.6 ± 2.2 h and 21.0 ± 2.1 h. The analysis of cumulative germination showed that all the lines had a longer lag phase in the presence of salt stress than in control conditions (Figure 2E, compare to Figure 2C).

### 2.3. The Severe Germination Delay of cue1-5 Is Caused by Stabilization of the Germination Repressor ABI5

The plant hormone abscisic acid (ABA) maintains dormancy and post-germinative seedling arrest under unfavorable environmental conditions [59]. ABA exerts these functions mainly through the basic leucine zipper (bZIP) transcription factor ABSCISIC ACID INSENSITIVE5 (ABI5), a key repressor of seed germination [60,61]. The antagonistic effects of ABA and NO during germination occur through a crosstalk during the regulation of *ABI5* transcription [62] and protein stability [45]. NO-mediated *S*-nitrosation of ABI5 at Cys 153 facilitates the degradation of this germination repressor, and it has been proven that ABI5 protein levels are high in NO-deficient mutant backgrounds [45]. As well as during germination, ABA has a prominent role during the regulation of the response to most abiotic stresses [63]. The regulation of seed germination and seedling establishment in response to stress is also regulated by ABI5, as it has been shown that loss-of-function *abi5* mutants were able to germinate and green even in the presence of 200 mM NaCl [60].

In order to confirm that the germination deficiency observed in the *cue1-5* mutant is indeed due to its lack of sufficient NO, we analyzed the protein levels of ABI5 in 4-day-old seedlings. In agreement with the marked NO deficiency of *cue1-5* and its inability to reach a similar G_max_ to that of the other lines, we found that this mutant accumulated higher ABI5 protein levels (Figure 3). The quantity of ABI5 was comparable in the rest of lines, in accordance with their similar NO levels. Confirming the role of NO in the stability of the protein, we also found equally elevated ABI5 levels in all the lines in the presence of high salinity in the Col-0 background, consistent with the presence of abiotic stress and the decreased NO levels and germination rate.

### 2.4. Early Root Elongation Is Impaired in cue1 Mutants Independently of Their NO Levels, but Root Cell Patterning Is Not Altered

Studies carried out mainly with pharmacological NO donors and scavengers show that both excessive and insufficient NO result in the inhibition of root elongation [32,33,64,65]. Analysis of the root length of 5-day-old seedlings showed that all the *cue1* mutants had shorter roots than their respective controls (Figure 4A). The root lengths of Col-0 and pOCA108 were equivalent (0.85 ± 0.04 cm and 0.77 ± 0.05 cm, respectively), despite their differences in DAF measurements of NO at 4 das. At the same time, the root lengths of *cue1-6* and *nox1* were 40% and 28% shorter than Col-0, respectively, even though both lines showed as much DAF fluorescence as Col-0. On the other hand, *cue1-1* and *cue1-5*, the mutants that accumulated less NO according to the DAF measurements, also had the shortest primary roots.

The root is organized into cell layers of different cell types that originate from a small set of cells at the core of the root, called the quiescent center (QC). QC cells divide infrequently and maintain the undifferentiated state of the adjacent cells, which are called stem cell initials. Stem cells continuously undergo asymmetric cell divisions that give rise to daughter cells that will divide symmetrically and start to differentiate [66]. This results in clear developmental zones along the longitudinal axis of the root. In the proximal meristem or meristematic zone (MZ), cells divide frequently until they are far enough from the stem cell niche. Once they reach the elongation/differentiation zone (EDZ) they start to expand quickly, beginning the process of differentiation. Maintaining these structures requires a balance between the generation of new cells in the meristem and the differentiation of cells in the EDZ. This balance determines the size of the root apical meristem (RAM) [67]. In accordance with their shorter roots, all the *cue1* mutants seemed to have a narrower vascular bundle than both wild-type lines, and a high accumulation of amyloplasts in the epidermis, cortex, and endodermis from the transition zone upwards (Figure 4B). A closer look at the meristematic zone showed that the altered organization of the stem cells around the QC that has been described to be caused by NO [32] was not apparent at this developmental stage (Figure 4B). This agrees with the fact that *cue1* mutants do not overproduce NO 4 days after completion of stratification.

### 2.5. Sugar Supplementation Modifies NO Production

In the absence of *CUE1*, plants are unable to establish photoautotrophic growth if they are not supplemented with exogenous metabolizable sugars, such as sucrose or glucose [24]. Chloroplasts are an important node of NO production [68], so we wanted to explore whether sugars have a role in NO homeostasis, since there are some reports that point to a possible role for NO in the regulation of energy production [69,70,71]. We tested the NO levels of *cue1-5*, *cue1-1*, and their respective controls after growing them for 7 das in a medium supplemented with either 2%(*w/v*) glucose or 0.75%(*w/v*) sucrose. Our results show that *cue1* mutants do indeed accumulate NO at this developmental stage (Figure 5A, Appendix A).

While the differences in NO levels are greater in the presence of glucose, the mutants also overproduce NO when grown with sucrose as a carbon source. *cue1-5* was the allele which exhibited greater DAF fluorescence, with 2.27 ± 0.05 a.f.u./μg protein in glucose and 1.48 ± 0.01 a.f.u./μg protein in sucrose. Compared to its control Col-0, this meant a 46% increase in NO content when grown with glucose, and a 23% increase in the presence of sucrose. *cue1-1* also showed a significant enhancement in NO levels compared to its control pOCA108, with 1.81 ± 0.14 a.f.u./μg protein in glucose compared to 1.49 ± 0.04 a.f.u./μg protein (18% increase), and 1.41 ± 0.04 a.f.u./μg protein in sucrose compared to 1.19 ± 0.03 a.f.u./μg protein (16% increase).

Since the mutants present an increased NO content in these growth conditions, we decided to measure primary root length to check whether there was a link between the carbon source, NO content, and root length in *cue1* mutants. We found that the root length of *cue1-5* and *cue1-1* was indeed significantly shorter (Figure 5B, Appendix A). After 7 days of growth, *cue1-5* had a 1.2 ± 0.06 cm root in sucrose, which meant a 32% reduction compared to Col-0, with 1.78 ± 0.05 cm. The reduction was even more apparent in glucose, where the primary root of *cue1-5* only elongated to 0.86 ± 0.09 cm, a 50% reduction compared to Col-0, with a 1.73 ± 0.04 cm root. Primary root growth was also inhibited in *cue1-1* with respect to pOCA108, in sucrose (1.10 ± 0.06 cm compared to 1.85 ± 0.06 cm, 41% reduction) as well as in glucose (1.01 ± 0.07 cm compared to 2.20 ± 0.07 cm, 54% reduction). Previous studies have suggested that an optimal concentration of NO is needed for proper root development, and that both excessive and deficient NO levels are detrimental for the plant [32,33], which coincides with our results. Interestingly, the NO content of *cue1-5* and *cue1-1* in sucrose was similar to the NO content of pOCA108 in glucose, but the root length of the *cue1* mutants was vastly different to that of pOCA108, which suggests that both NO content and the sugar available as a carbon source have a role in the modulation of primary root growth.

We also explored the primary root apical meristems of these roots to analyze meristematic cortical cell length, meristem size, and meristem cell number of the mutants (Figure 6, Appendix A). Morphological observation showed that all the lines accumulated amyloplasts in the cortical and epidermal cells upwards of the transition zone when they were grown in the presence of glucose (Figure 6A), but not in the presence of sucrose, where only *cue1-5* and *cue1-1* accumulated them (Figure 6B). Furthermore, we noticed that the *cue1* mutants presented amyloplasts in the columella stem cell layer, an indication of earlier differentiation. As described in [32], we observed disorganization of the cells surrounding the QC in some of the *cue1-1* and *cue1-5* plants, especially in the presence of glucose, where the endogenous NO levels were higher.

An analysis of root cortical cells showed that the differences among genotypes were clearer when the plants had been grown with glucose as a carbon source. In those conditions, *cue1-5* and *cue1-1* differentiated earlier than their controls, since the EDZ occurred closer to the initial cells than in their respective wild-type controls. While a statistical analysis showed that the carbon source only caused a significant difference in meristem size in *cue1-5* seedlings (Figure 6C), the variations observed in the rest of the lines supported the divergences observed in the root length of the mutants. Thus, the RAM size in *cue1-5* was 21% smaller than its control in glucose and 18% smaller in sucrose, while *cue1-1* presented only a 2% reduction in sucrose, but a 15% decrease in glucose compared to pOCA108. The same was true for the number of cortical cells in the apical meristem, as both *cue1-5* and *cue1-1* had significantly fewer cells than their respective controls in glucose, but not in sucrose (Figure 6D). This decrease in meristematic cell number was compensated by an increase in meristematic cell length (Figure 6E). These observations support the existence of an interplay between NO content and sugar metabolism in the modulation of primary root growth. At the same time, these results have allowed us to identify *cue1* mutants as suitable NO-overproducer plant lines to explore the role of NO during root growth.

## 3. Discussion

### 3.1. CUE1 Does Not Directly Contribute to the Rapid NO Synthesis during Seed Imbibition

NO is a stimulator molecule in plant photomorphogenesis, as it promotes seed germination and de-etiolation, and inhibits hypocotyl and internode elongation [42]. It has been demonstrated that NO-deficient mutants exhibit increased dormancy, hypersensitivity to ABA during seed germination and seedling establishment, as well as resistance to dehydration [72,73]. The decrease in NO during germination observed in *cue1-5* induces a general decline in germination parameters, since it presents a reduced maximum germination rate, decreased uniformity of germination among seeds of the same population, a delay in the initiation of germination and slower germination overall. Moreover, the rest of *cue1* mutants were unable to accumulate more NO than their controls at this developmental stage. It has been previously demonstrated that a NO burst is required for proper germination and post-germinative growth [12,50,74], possibly to counteract the inhibitory effect of the ABA-regulated transcription factor ABI5 [45,62,75], which acts as the main regulator of one of the earliest developmental checkpoints to spare the plant from pouring resources into growth when the environmental conditions are not optimal for the development of the seedling [60,61].

During germination, when the quiescent seed reactivates its metabolism, there is a surge in the production of NO that slows down three hours after imbibition [50,74], although NO synthesis can be detected in the aleurone layers as early as fifteen seconds after exogenous nitrate addition [12]. It is thought that the generation of NO by the seed is non-enzymatic because of its quick response, and that this non-enzymatic synthesis requires acidic pH and the presence of compounds that can act as antioxidants, so this synthesis pathway would be restricted to the apoplast of aleurone cells and maybe local areas of the root during transient acidification caused by alteration of nutrient supply. The aleurone layer fulfills both requisites, as its pH is usually between 3 and 4, and its plastids contain proanthocyanidins, phenolic compounds with antioxidant capacity [12]. The early stages of germination coincide with a depletion of oxygen, so it is assumed that seeds are in an anaerobic state until the radicle breaks through the testa, when oxygen gradually returns the seed to aerobic conditions [76]. Thus, it is highly unlikely that the seed can offer the oxidative environment necessary to obtain NO from arginine, and it would explain why *cue1* mutants, which are thought to be involved in the oxidative biosynthesis of NO, do not overproduce NO during germination.

Among all studied *cue1* mutants, only *cue1-5* mutants do not germinate as well as wild-type seeds in response to advantageous environmental conditions even after dormancy is broken by stratification. Our results show that this is due to its inability to de-stabilize ABI5, as shown by its increased ABI5 protein levels. The increase in ABI5 is a direct consequence of the reduced endogenous NO content of *cue1-5*. However, we do not think this is directly caused by the loss of function of *CUE1*, as the other alleles behave closer to their wild-type controls, both in endogenous NO levels and germination parameters. This mutant line contains an additional *transparent testa* mutation that affects seed pigmentation, testa solidity and germination [27].

Interestingly, data from the Arabidopsis Seed Coat eFP Browser [77] and the ePlant Browser [78] indicate that the expression of *CUE1* in the mature embryo and in dry seeds is almost negligible, but it increases steadily one day after imbibition, well after the rapid production of NO that starts germination. On the other hand, it is highly expressed in the ovaries of the plant, which will give rise to the seed coats. The expression of *CUE1* increases in the developing embryo until the walking stick stage, 7–8 days post-anthesis, at the end of the cellularization of the endosperm and before the accumulation of reserves. It is possible, then, that its role in NO generation during germination is carried out during embryo development and seed maturation, likely in an indirect fashion. Since the non-enzymatic synthesis of NO requires proanthocyanidins [12], which are generated as one of the end products of the shikimate pathway, it could be expected that *cue1* seeds have less of these polyphenolic compounds available for the quick apoplastic formation of NO and thus a longer lag phase. Indeed, *cue1* mutants have reduced flavonoids, hydroxycinnamic acids, and simple phenolics [27], in agreement with this hypothesis.

### 3.2. High Salinity Impairs Germination by Increasing ABI5 and Decreasing NO Levels

Plants respond to abiotic stress by increasing the production of ABA and reactive oxygen and nitrogen species, so we also tested the behavior of *cue1* mutants in response to salt stress during early plant development. Depending on its concentration, NO can protect plants against salt stress by lessening the secondary oxidative stress induced by high salinity [46,47,79,80], or it can enhance sensitivity to the stress if its accumulation is excessive, causing additional nitrosative stress [36,81,82]. It is thought that the protective role of NO is exerted mainly through redox modification or *S*-nitrosation of ROS scavenging enzymes, antioxidant systems, and respiratory pathways [83]. Reports on the accumulation of NO in *A. thaliana* in response to high salinity are sometimes contradictory, possibly because of differences in biological material, the extent of the stress, and the developmental stage at which the plant was subjected to salt stress [46,47,53,54,55,56,84,85].

In our growth conditions and developmental stage, the NO content of the seedlings decreased when seeds had been germinated on MS medium supplemented with 100 mM NaCl, regardless of their genotype and in agreement with [46]. Contrary to other salt stress treatments performed in *A. thaliana*, this may not be enough to cause secondary nitrosative stress. It has been demonstrated that the induction of NO by salt requires peroxisomal NO synthase activity [53], which needs oxygen to be able to oxidize arginine. Thus, it is possible that, in the anoxic state of the germinating seed, 100 mM NaCl is too mild a stress to generate enough ROS for creating the oxidative environment required for the function of NO synthase. Because of the reduced NO levels, germination was equally impaired in all the lines, as evidenced by the analysis of different germination parameters. This was explained by an increase in ABI5 protein levels, which was easily detected after germination during salt stress, since ABI5 is stabilized by ABA and degraded in a NO-mediated process [33], and it accumulates in seeds that undergo salt stress [60]. In fact, ABI5 expression is highly induced by abiotic stress at the transition from mature seeds to seedling growth. Salt delays the decline of ABI5 levels and promotes its expression throughout the seedling, while in unstressed conditions it would be undetectable [86].

### 3.3. Initial Seedling Establishment Is Impaired in cue1 Mutants Independently of Their NO Levels

In addition to the characterization of the role of *CUE1* during germination, we explored its role during early post-germinative growth by quantifying the primary root length and exploring the root apical meristem of *cue1* mutants 5 das. However, we were unable to find a relationship between their NO content and these parameters, since all *cue1* lines showed a significant reduction in root growth independently of the differences in their NO levels. This observation further stresses the need for a careful assessment of NO levels when working with NO mutants. Understanding the effect of the loss of a translocator involved in glycolysis and photosynthesis during root development can prove to be quite complex, as phenotypes are the result of an intricate interplay between hormones, other growth regulators, and environmental cues.

Early plant development is mostly supported by the storage reserves found in the endosperm [87], which in *A. thaliana* cannot last more than 4-5 days and are mostly used to elongate the hypocotyl until it reaches light, when cotyledons start greening and are converted into photosynthetic organs [88]. *cue1* mutants have been described to be defective in the maturation of eoplasts to chloroplasts and in the initial establishment of photoautotrophism in the absence of an exogenously supplied metabolizable sugar [24]. Sugars mobilized from the endosperm and synthesized in the green parts of the plant function as energy sources as well as signaling molecules. In particular, photosynthetic sugars delivered from the cotyledons to the root act as interorgan signals to initiate root growth and have a dominant role during the cotyledon stage of seedling development, even over the phytohormone auxin, which is thought to be essential for the regulation of root development [88]. Since all *cue1* mutants are photosynthetically defective, we cannot rule out that their initial inability to start root elongation stems from the lack of a photosynthetic signal.

### 3.4. Sugars Alter Root Development through Modulation of NO Homeostasis

After exposure to exogenous NO, the content of several glycolysis intermediates, metabolites of the TCA cycle and intermediates of the Calvin cycle were reduced [70]. On the other hand, the content of sucrose and different monosaccharides, disaccharides, amino and nucleotide sugars increased [70,71]. These metabolic changes are caused by the transcriptional upregulation of glycolytic enzymes and the downregulation of photosynthetic proteins [69], and by the regulation via *S*-nitrosation of enzymes involved in sugar metabolism, such as ATP synthase, enolase, or phosphoglycerate kinase [71].

*CUE1/PPT1* has a central role in sugar metabolism because it is the only source of PEP into the chloroplast and its absence directly impacts carbon partitioning [27,89]. Interestingly, *cue1* mutants are unable to establish photoautotrophic growth right after germination and need to be exogenously supplemented with a fixed carbon photoassimilate [24]. These mutants have impaired light signaling and a reduced capacity of de-etiolation [24,26], possibly because they are defective in chloroplast maturation [24]. Essentially, *cue1* mutants behave as heterotrophs during early plant development.

*cue1* mutants have not been the first NO mutants to be linked to sugar metabolism. *noa1*, which encodes a cGTPase necessary for assembling plastid ribosomes [90], was initially isolated as a mutant with less endogenous NO content and essential for its production [91]. Interestingly, the levels of NO in *noa1* mutants can be partially recovered by exogenous addition of sucrose [92], and the mutant presents chloroplast biogenesis defects [93] and reduced fumarate, even though its energy status and redox potential seem unaffected [92]. The NO deficiency in *noa1* was explained as an indirect effect of its reduced ability to generate photosynthates [93]. However, *cue1* mutants are also defective in chloroplast biogenesis and contain fewer photoassimilates than wild-type lines, whereas their NO levels are enhanced. Thus, a reduced photosynthetic capacity cannot solely be the reason for reduced NO content in *noa1*.

We analyzed the endogenous NO status of *cue1* mutants in the presence of two different sugars that can be used by the plant as a source of reduced carbon and energy, glucose and sucrose. If a general decrease in reduced carbon availability explained the alteration in NO content, the mutants would be expected to produce the same NO in any case. However, our results showed the differences between *cue1* mutants and their controls were greater when the seedlings had been grown in the presence of glucose. Sucrose did not revert the NO overproducing phenotype of *cue1* plants, but it did diminish the differences among the lines, as in the case of *noa1* [93]. The increased NO production in the presence of glucose in *cue1-5* and *cue1-1* caused a decrease in primary root elongation rate, meristem size, meristematic cell number, and longer root cell length, corroborating the findings of [29]. These differences were not as obvious when the plants were grown in the presence of sucrose, but the root growth of *cue1-5* and *cue1-1* was still affected. Our findings show that NO accumulation depends on sugar metabolism. Nitrate reductase (NR) is an enzyme that also participates in the production of NO [94] and it has been previously shown that the expression of *NIA2*, the gene encoding NR, is light-induced, while NR activity is linked to photosynthesis [95]. NR could be a good candidate linking NO homeostasis and carbon metabolism, but, surprisingly, *noa1* has higher NR levels [92], while *cue* mutants have reduced NR activity [26]. Understanding the role of carbon metabolism will help elucidate the molecular mechanisms underlying NO production in plant development, but further experiments are required.

## 4. Materials and Methods

### 4.1. Plant Lines

*Arabidopsis thaliana cue1-5* [26], *cue1-6* [27] and *nox1-1* [23] plants are in the Columbia (Col-0) ecotype background, while *cue1-1* [24] is in the Bensheim (Be-0) ecotype background. All *cue1* mutants are defective in the *CUE1* locus (AT5G33320). The *cue1-5* (CS3156) and *cue1-6* (CS3168) alleles were generated by mutagenizing a Col-0 population with ethyl methanesulfonate (EMS) [27], while the *nox1-1* allele was generated by fast neutron mutagenesis [23]. Additionally, *cue1-5* seeds are yellow and lack brown pigments in the seed coat, as the mutant contains an additional *transparent testa*/*glabrous* mutation [96]. The *cue1-1* allele was generated by mutagenizing a population of line pOCA108-1 with gamma radiation [24]. pOCA108-1 is a single-insertion line that contains the reporter construct pOCA108 on chromosome 2. This construct contains the alcohol dehydrogenase (*ADH*) gene under the control of chlorophyll *a/b* binding protein (*CAB3*) promoter, and was transformed into Bensheim line R002, which contains a null mutation in the endogenous *ADH* gene [24].

### 4.2. Plant Growth Conditions

Seeds grown in vitro were surface sterilized using a bleach solution (25% bleach, 0.1% Tween). Sterilized seeds were stratified in water for 48–72 h at 4 °C to help synchronize germination. Stratified seeds used for root elongation assays and NO quantification in different carbon sources were grown on plates containing a modified Murashige and Skoog (MS) medium [97] optimized for root growth, MS-Root [2.3 g/L MS (Duchefa Biochemie, Haarlem, The Netherlands), 15 g/L agar], supplemented with either sucrose or glucose as indicated. The greater agar content allows for vertical growth of seedlings on the surface of the medium. For the rest of the experiments, plants were grown on plates containing 4.9 g/L MS, 2% glucose, 6 g/L agar. Stratified seeds used for NO quantification, germination and high salinity assays were grown on plates containing MS medium supplemented with 2% glucose. To analyze sensitivity to salt stress, MS was supplemented with 100 mM NaCl (PanReac AppliChem, Darmstadt, Germany). Plants were grown under a 16 h light/8 h dark photoperiod at a constant temperature of 21 °C and 50–60% humidity.

### 4.3. Germination Assays

Germination was determined as radicle emergence at indicated times. The analysis of germination parameters was carried out using the GERMINATOR software [58]. The parameters used are t_50_, the time it takes for 50% of the viable seeds to germinate, for speed of germination, and U_7525_, the elapsed time from 25% to 75% of the viable seeds to germinate, for uniformity of germination. Germination was also represented with a time course graph as cumulative germination.

### 4.4. Western Blotting

Total proteins were extracted from 4-day-old stratified and imbibed seeds (500–600 per genotype) for western blot analysis. Tissue was homogenized using a Silamat S6 homogenizer (Ivoclar Vivadent, Madrid, Spain) until all tissue was completely powdered. Samples were incubated with an extraction buffer containing 100 mM Tris-HCl, 150 mM NaCl, 0.25% NP-40 and 1× cOmplete EDTA-free Protease Inhibitor Cocktail (Sigma, Saint Louis, MO, USA). Protein concentration was determined by the Bio-Rad Protein Assay (Bio-Rad, Hercules, CA, USA) based on the Bradford method [98]. An amount of 90 µg of total protein was loaded per well in 10% SDS-acrylamide/bisacrylamide gel electrophoresis using Tris–glycine–SDS buffer. Proteins were electrophoretically transferred to an Immobilon-P polyvinylidene difluoride membrane (Merck Millipore, Burlington, VT, USA) using the semi-dry Trans-Blot Turbo Transfer system (Bio-Rad). Membranes were blocked in Tris-buffered saline-0.1% Tween 20 containing 5% Blocking Agent and probed with antibodies diluted in blocking buffer. Anti-ABI5 Purified Rabbit Immunoglobulin (Biomedal, Sevilla, Spain, 1:10,000) and anti-Actin clone 10-B3 Purified Mouse Immunoglobulin (Sigma-Aldrich, A0480, 1:10,000) antibodies were used in the Western blot analyses. Detection was performed using ECL Advance Western Blotting Detection Kit (Amersham, Chicago, IL, USA) and the chemiluminescence was detected using an Intelligent Dark-Box II, LAS-1000 scanning system (Fujifilm, Tokyo, Japan).

### 4.5. Detection of NO Production

Freshly prepared protein extracts prepared as for western blotting were used to assay NO content. An amount of 20 μL of each protein extract was incubated with 180 μL of a solution containing 10 μM of DAF-FM DA (Sigma-Aldrich) in 50 mM HEPES buffer pH 7.5 in microtiter plates, following the method described in [49]. Samples were incubated at 37 °C for 2 h in the dark. After incubation, the emitted fluorescence of each well was measured in a Varioskan LUX Multimode Microplate Reader (ThermoFisher Scientific, Waltham, MA, USA). Samples were normalized by their total protein content (Appendix A) and against a control condition in each experiment. Blanks included in all experiments behaved similarly and emitted a negligible signal, that was subtracted from all experimental samples.

Detection of NO through confocal microscopy was performed using the same experimental conditions and DAF-FM DA staining protocol for the former spectrofluorometry measurements. λ scan 500–666 nm was used to set the emission window and FIRE LUT to represent a fluorescence heatmap of intensity.

### 4.6. Root Growth Analysis

After full germination, root growth was captured by scanning plates with an Epson flatbed scanner and a resolution set to 600 ppi. Primary root length of individual seedlings was then measured using Fiji [99]. Average root elongation rate (mm/d) was calculated as an average of daily root elongation rates following the protocol described in [100].

Primary root apical meristems were analyzed to measure meristematic cortical cell length (μm), meristem size (μm), and meristem cell number by performing a propidium iodide (PI) stain following the protocol described in [101]. Root tips were examined using a Leica SP2 confocal microscope with a 40× oil immersion objective. The resulting image data was processed with semi-automatic image analysis software, Cell-O-Tape [102].

### 4.7. Statistical Analysis

For each dataset, the distribution was initially assessed by plotting all the values of the dependent variable as a histogram. Normality and homoscedasticity of the populations were determined using the Shapiro–Wilk and Levene’s tests, respectively. An appropriate statistical model was then selected depending on the number of independent variables, the distribution of the dependent variable, and whether it was categorical or continuous. To account for type I error, data was presented with a 95% confidence interval (CI). Generalized linear models (ANOVAs) and Pearson’s product-moment correlation tests were performed using R Statistical Software (R version 4.0.2, R Foundation for Statistical Computing) [103] in the RStudio environment (RStudio version 1.3.959, PBC) [104]. Excel (Microsoft Office 365 ProPlus, v.1902) was used for other statistical tests and graph plotting. The statistical power of the chosen tests was performed using G*Power v.3.1.9.2 (Franz Faul, Universität Kiel).

## 5. Conclusions

Despite the ample research performed to elucidate the role of NO during development, no systematic study of the production of NO during different developmental stages has been performed to date. The reports discussed in this article point to specific roles that would require tightly controlled spatio-temporal NO accumulation. To our knowledge, the only analysis comparing the production of NO at two different plant developmental stages was published using *Medicago truncatula* [105]. In this study it was shown that senescing plants had an increased sensitivity to nitrosative stress, as well as repression of nitrate uptake and NR activity, suggesting that accumulation of NO and regulation of its homeostasis depends on the developmental stage. Our results support this statement, as a careful characterization of the NO production of different *cue1* mutants, routinely used as NO overproducer mutants, proved that *cue1* mutants do not accumulate NO during early plant development, but they do at later stages. Since most NO mutants are defective in proteins involved in primary metabolism [68,106], we recommend that NO quantification be performed for NO mutants at early developmental stages, given that their alteration of NO homeostasis might stem from unexpected effects of their mutations. In conclusion, our results demonstrate that *cue1* is a useful tool to study the physiological functions of NO, since this mutant accumulates NO under controlled experimental conditions that require awareness of the developmental stage and growth conditions of the plants, especially in terms of stress trade-off.

## Figures and Tables

**Figure 1 plants-09-01484-f001:**
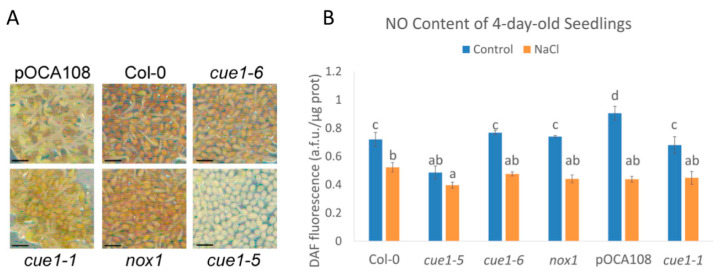
Phenotype (**A**) and nitric oxide (NO) content (**B**) of 4-day-old seedlings. Scale bar, 1 mm. DAF fluorescence intensity of extracts of 4-day-old seedlings grown on Murashige and Skoog (MS) with and without 100 mM NaCl. Values represent the mean ± CI (*n* = 3). Bars with common letters (a–d) do not show significant statistical differences. A two-way analysis of variance (ANOVA) with post-hoc Tukey’s honestly significance difference (HSD) test showed a statistically significant interaction between genotype and salinity ((F_(5,24)_ = 23.13, *p* < 0.001, eta2[g] = 0.83). An analysis of simple main effects for each factor was performed with statistical significance after a Bonferroni correction (Appendix A).

**Figure 2 plants-09-01484-f002:**
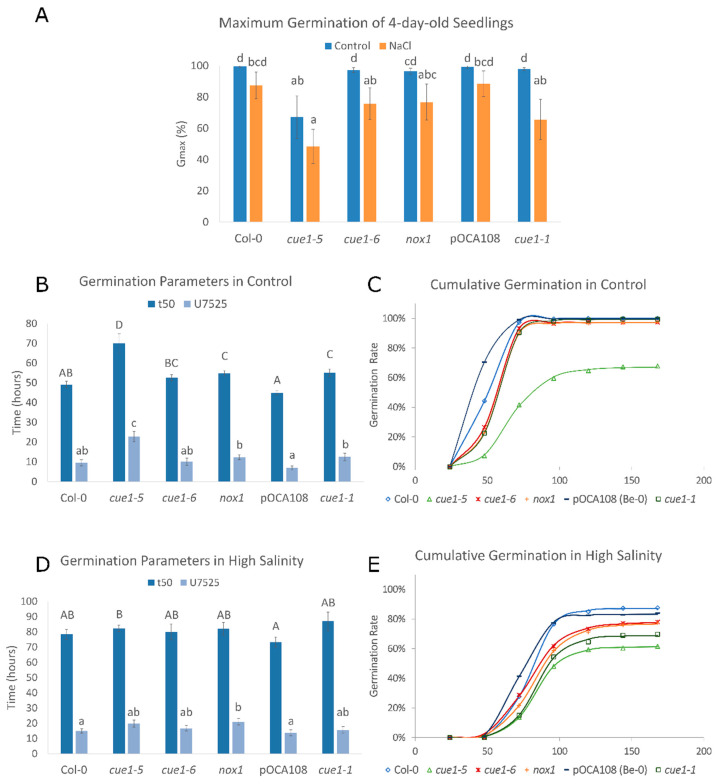
NO is required for germination vigor. Values represent the mean ± CI (*n* = 12). Bars with common letters do not show significant statistical differences as determined by the respective statistical tests: (**A**) Maximum germination of seedlings 4 das with and without 100 mM NaCl. Welch’s one-way ANOVA with post-hoc Games–Howell test was F_(5,29)_ = 7.98. *p* < 0.001. (**B**) Germination parameters t_50_ and U_7525_ for seeds germinated in control conditions. (**C**) Representative cumulative germination curve for seeds germinated in control conditions as predicted by GERMINATOR. (**D**) Germination parameters t_50_ and U_7525_ for seeds exposed to high salinity. (**E**) Representative cumulative germination curve for seeds germinated in the presence of 100 mM NaCl as predicted by GERMINATOR.

**Figure 3 plants-09-01484-f003:**
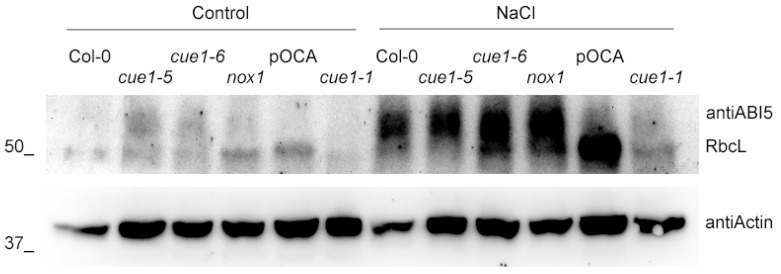
Germination inhibitor ABI5 is stabilized in *cue1-5* and in the presence of salt stress. Western blot analysis of ABI5 accumulation in the different *cue1* lines grown on MS with and without 100 mM NaCl for 4 das. Actin levels are shown as loading controls.

**Figure 4 plants-09-01484-f004:**
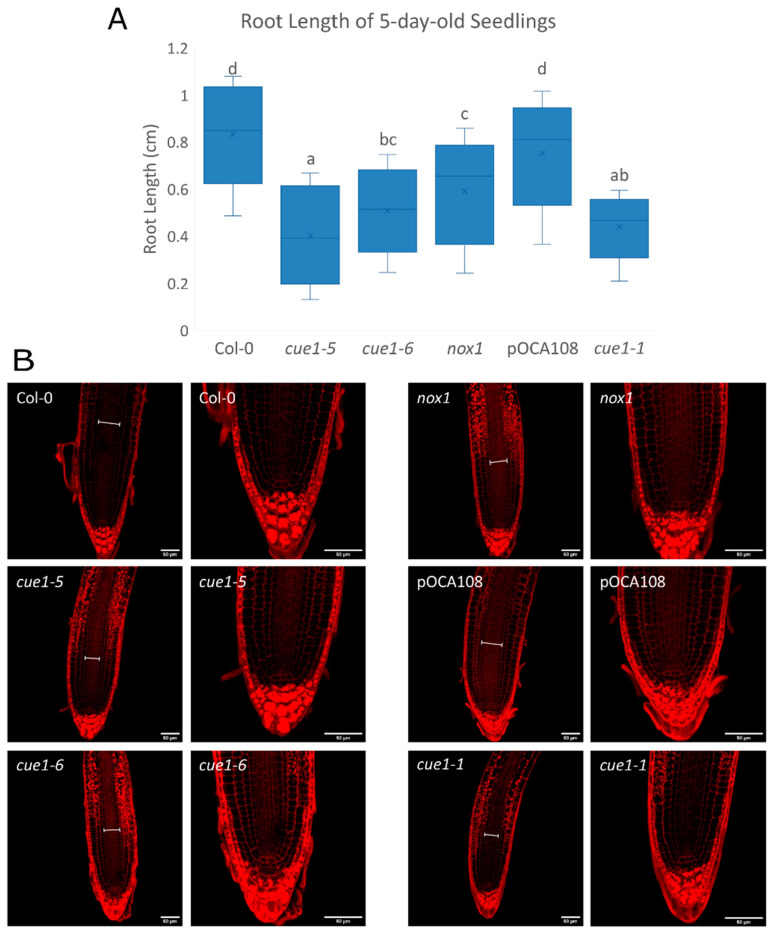
Root elongation is defective in all *cue1* mutants. (**A**) Primary root length of 5-day-old seedlings. The diagram shows data between the lower (Q1) and upper (Q3) quartiles, the median and the mean (x) for each genotype. Bars with common letters (a–d) do not show significant statistical differences as determined by one-way ANOVA with post-hoc Tukey’s HSD test (F_(5,214)_ = 57.8, *p* < 0.001). (**B**) Representative images of the root apex of 5-day-old seedlings stained with Schiff propidium iodide. Scale bars on the bottom right corner correspond to 50 μm. The width of the vascular bundle is indicated with a white line in each root.

**Figure 5 plants-09-01484-f005:**
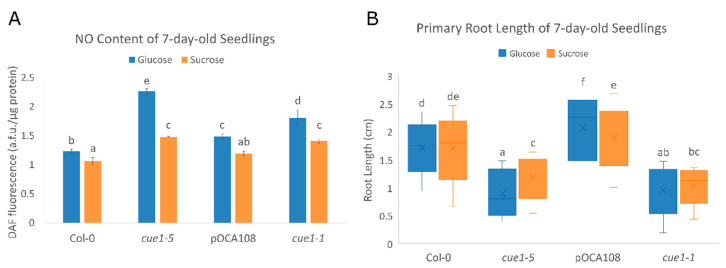
(**A**) The NO content of *cue1* alleles depends on their developmental stage. DAF fluorescence intensity of extracts of 7-day-old seedlings grown on MS-Root medium supplemented with either 2%(*w/v*) glucose or 0.75%(*w/v*) sucrose. Values represent the mean ± CI (*n* = 4). A two-way ANOVA with post-hoc Tukey’s HSD test showed a statistically significant interaction between genotype and carbon source ((F_(3,24)_ = 33.99, *p* < 0.001, eta2[g] = 0.81). An analysis of simple main effects for each factor was performed with statistical significance after a Bonferroni correction (Appendix A). (**B**) Primary root length is influenced by NO and sugar. Primary root length of 7-day-old seedlings grown on MS-Root supplemented with either 2%(*w/v*) glucose or 0.75%(*w/v*) sucrose (*n* = 50). The diagram shows data between the lower (Q1) and upper (Q3) quartiles, the median and the mean (x) for each genotype. Common letters (a–f) indicate there are no significant statistical differences as determined by Welch’s one-way ANOVA with post-hoc Games–Howell test (F_(7,571)_ = 203.57, *p* < 0.001).

**Figure 6 plants-09-01484-f006:**
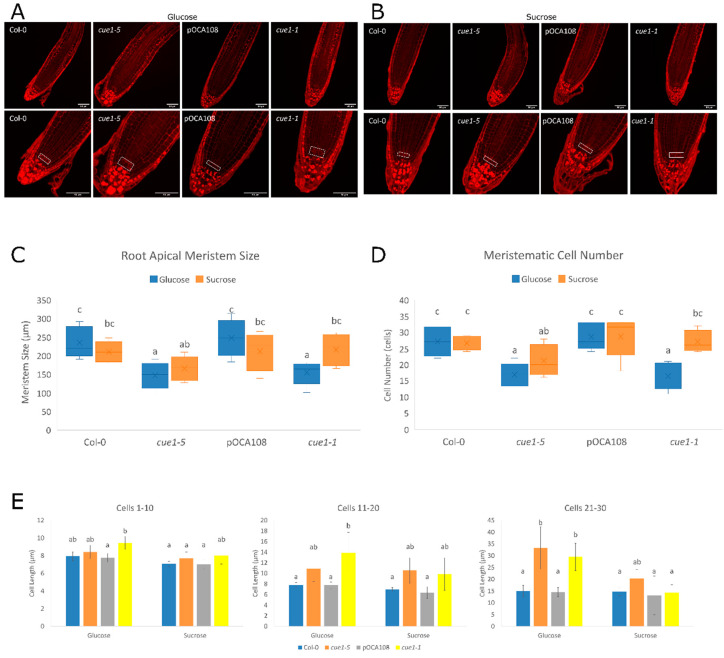
Increased NO causes disorganization of root meristematic stem cells. (**A**) Representative images of the root meristem of 7-day-old seedlings grown on MS-Root medium supplemented with 2%(*w/v*) glucose. (**B**) Representative images of the root meristem of 7-day-old seedlings grown on MS-Root medium supplemented with 0.75%(*w/v*) sucrose. Seedlings were stained with Schiff propidium iodide. Scale bars on bottom left corner correspond to 50 μm. A white rectangle was drawn around the stem cells. Differentiation parameters were quantified from these images. Statistical differences as determined by two-way ANOVA with post-hoc Tukey’s HSD test are represented by different letters (a–c). (**C**) Meristematic size (F_(3,70)_ = 6.65, *p* < 0.001, eta2[g] = 0.22). An analysis of simple main effects for each factor was performed with statistical significance after a Bonferroni correction (Appendix A). (**D**) Number of cells in the root apical meristem (F_(3,70)_ = 6.16, *p* < 0.001, eta2[g] = 0.21). An analysis of simple main effects for each factor was performed with statistical significance after a Bonferroni correction (Appendix A). (**E**) Bars with common letters (a–c) are not significantly different according to Welch’s ANOVA with post-hoc Tukey’s HSD test. Average cortical cell sizes are shown for cortical cells 1–10 (F_(7,73)_ = 3.99, *p* < 0.001), 11–20 (F_(7,73)_ = 5.45, *p* < 0.001) and 21–30 (F_(7,73)_ = 7.23, *p* < 0.001) counted from the QC. Values represent the mean ± CI (*n* = 10).

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
