# Peer review of "Nitric Oxide Overproduction by cue1 Mutants Differs on Developmental Stages and Growth Conditions"

_plants, 2020, doi:10.3390/plants9111484_

Round 1

Reviewer 1 Report

Dear Authors,

This is a valuable manuscript regarding Nitric oxide signaling and their role in developmental process and abiotic stress. The manuscript is well written and result corroborates with previous research. Thus, I would like to recommend this manuscript for publication after minor revision comments below:

Figure 5 and 6: In both figures, resolution of image need to increase, since image looking blurred.

I would also like to see few references to recent studies. It would seem that studies that have been published last 2 years have not been included much references. Some recent suitable reference may be added.

Some recent reference are not included such as:

Rai KK, Pandey N, Rai SP. Salicylic acid and nitric oxide signaling in plant heat stress. Physiologia plantarum. 2020, 168(2):241-55.

Bruand C, Meilhoc E. Nitric oxide in plants: pro-or anti-senescence. Journal of experimental botany. 2019, 70(17):4419-27.

Astier J, Gross I, Durner J. Nitric oxide production in plants: an update. Journal of Experimental Botany. 2018, 69(14):3401-11.

In reference section: All scientific name should be in italic font. Such as references 15 and 29.

Author Response

This is a valuable manuscript regarding Nitric oxide signaling and their role in developmental process and abiotic stress. The manuscript is well written and result corroborates with previous research. Thus, I would like to recommend this manuscript for publication after minor revision comments below:

We thank the reviewer for reserving some of their valuable time to our manuscript. We are grateful for the constructive comments and have adapted the manuscript according to his/her suggestions.

Below, we include a point by point overview to the reviewer's comments.

Figure 5 and 6: In both figures, resolution of image need to increase, since image looking blurred.

We appreciate the critical look at the Figures and we apologize for their quality, once more due to the file compression to PDF. We have submitted vector images of each these figures along with the manuscript.

I would also like to see few references to recent studies. It would seem that studies that have been published last 2 years have not been included much references. Some recent suitable reference may be added.

Some recent reference are not included such as:

Rai KK, Pandey N, Rai SP. Salicylic acid and nitric oxide signaling in plant heat stress. Physiologia plantarum. 2020, 168(2):241-55.

Bruand C, Meilhoc E. Nitric oxide in plants: pro-or anti-senescence. Journal of experimental botany. 2019, 70(17):4419-27.

Astier J, Gross I, Durner J. Nitric oxide production in plants: an update. Journal of Experimental Botany. 2018, 69(14):3401-11.

We have updated our references to include all of these suggestions.

In reference section: All scientific name should be in italic font. Such as references 15 and 29.

We have included italics in all scientific names.

Reviewer 2 Report

The manuscript characterizes NO levels and developmental traits of cue1 mutants of Arabidopsis thaliana, concluding that the expected NO overproduction phenotype occurs only at specific developmental stage. This is a very relevant topic for NO studies in plants and the authors did a good job in the discussion of their data. However, there are some important aspects that requires improvements.

1 - The abstract has long introductory and method sections, but the results are overlooked. The authors should revise the abstract in order to explore more the results of the study.

2 - Lines 33-34: Mammalian cells can also produce NO from nitrite. Please revise this phrase.

3 – Please define all abbreviations in their first mention (e.g., ROS in line 43).

4 – Lines 52-55: I cannot agree that the reported half-lives of S-nitrosothiols are as long as claimed here (years in water?). You can check in several studies that evaluate the NO release from GSNO in water, for example, that the its half-life is hours and can even be minutes depending on the environmental conditions (as light, pH, presence of copper, temperature). So, I am not convinced that “the spontaneous homolytic cleavage of the S-NO bond to release NO is improbable in planta.” Please revise this part checking more recent literature. However, I completely agree with the other arguments about the limitations of NO donors (not  always replicate the endogenous effects of NO; SNP is in fact a nitrosonium cation donor that also generates cyanide; the application of NO donors might result in nitrosative stress; there have been very few attempts to understand the kinetics of NO generation in planta by NO donors). Perhaps you could say that, in addition to release NO, S-nitrosothiols can act through trans-S-nitrosation reactions.

5 – Lines 118-126: I am not sure if this basic explanation is needed here (as well as all other in the Results section).

6 – Are there changes in the protein content among the mutants and WT? I am asking it as NO production could not be standardized by protein content if this parameter differ among the genotypes. I suggest to show protein content as supplemental material to avoid this criticism.

7 – Overall, the figure formatting requires improvements. There are Y-axis without line, unnecessary grid lines, number with commas instead of points, etc.).

8 – When comparing the genotypes and salt stress (2 factors), I think that a two-way ANOVA would be more adequate than the one-way ANOVA used. It also necessary when comparing the genotypes and carbon source.

9 – Lines 198-199: A statistical analysis of correlation (as linear regression, Pearson’s correlation, etc.) is required to conclude that there is a correlation between the variables. This valid for most of the data of the manuscript to demonstrate that a phenotypic variable (as germination, root lengths, etc.) is indeed correlated to NO level.

10 – Line 255: It is not the case of all mutants, as no difference was detected between WT an pOCA108.

11 – Lines 297-298: It is essential to check whether the higher NO levels of cue1 mutants also occur without the carbon supplementation, otherwise they cannot claim that it is related to the developmental stage (it can be related solely to the sugar treatment). Also, the control without sugar supplementation is required in Figure 6. Or in the other experiments (Figures 1-4), have the medium received carbon supplementation? Please clarify this crucial point.

12 – In Figures 5 and 6, I really missed the two mutants that were not analyzed. They should be included here or removed from the whole manuscript, otherwise the discussion and conclusions become confusing and cannot be generalized.

13 – Also in Figures 5 and 6, the use of cPTIO is necessary to demonstrate that NO level is indeed involved in the observed developmental phenotypes and that they are not related to other modifications that occur in such pleiotropic mutants.

14 – Lines 371-378: This information has been provided before in the text. It is repetitive.

15 – Line 383: It is better to use “decrease” rather than “lack” of NO. For me, lack sounds as if there was no NO in the mutants.

16 – The mutants have different ecotype backgrounds. However, only Col-0 was used as a WT control. So, how the authors can ensure that the differences between pOCA108-1/cue1-1 and Col-0 are not related to the different ecotypes?

17 – NO production was quantified spectrofluorometry using protein extracts, which is not a good procedure. In this condition, tissues and probably organelles are not intact so that many artifacts can occur in DAF fluorescence. A negative control with cPTIO is necessary in this assay (as done in microscopy analysis), particularly in those in Figure 1 (there is no microscopy analysis to compare).

Author Response

The manuscript characterizes NO levels and developmental traits of cue1 mutants of Arabidopsis thaliana, concluding that the expected NO overproduction phenotype occurs only at specific developmental stage. This is a very relevant topic for NO studies in plants and the authors did a good job in the discussion of their data. However, there are some important aspects that requires improvements.

We thank the reviewer for reserving some of their valuable time to our manuscript. We are grateful for the constructive comments and have adapted the manuscript according to his/her suggestions.

Below, we include a point by point overview to the reviewer's comments.

1 - The abstract has long introductory and method sections, but the results are overlooked. The authors should revise the abstract in order to explore more the results of the study.

Thank you for noticing that our abstract was not concrete enough. Abstract has been fully modified according to the suggestions of the reviewer.

2 - Lines 33-34: Mammalian cells can also produce NO from nitrite. Please revise this phrase.

This line has been corrected to make our statement clearer.

3 – Please define all abbreviations in their first mention (e.g., ROS in line 43).

All abbreviations have been defined in their first mention.

4 – Lines 52-55: I cannot agree that the reported half-lives of S-nitrosothiols are as long as claimed here (years in water?). You can check in several studies that evaluate the NO release from GSNO in water, for example, that the its half-life is hours and can even be minutes depending on the environmental conditions (as light, pH, presence of copper, temperature). So, I am not convinced that “the spontaneous homolytic cleavage of the S-NO bond to release NO is improbable in planta.” Please revise this part checking more recent literature. However, I completely agree with the other arguments about the limitations of NO donors (not  always replicate the endogenous effects of NO; SNP is in fact a nitrosonium cation donor that also generates cyanide; the application of NO donors might result in nitrosative stress; there have been very few attempts to understand the kinetics of NO generation in planta by NO donors). Perhaps you could say that, in addition to release NO, S-nitrosothiols can act through trans-S-nitrosation reactions.

Thank you for the comment; we realize that it is difficult to ascertain exactly how and when the homolytic cleavage of the S-NO bond occurs in planta. The text has been removed and modified to reflect the reviewer`s highlights.

5 – Lines 118-126: I am not sure if this basic explanation is needed here (as well as all other in the Results section).

We appreciate this suggestion. We have removed basic explanations and think in this revised version the Results section has been described in a suitable way.

6 – Are there changes in the protein content among the mutants and WT? I am asking it as NO production could not be standardized by protein content if this parameter differ among the genotypes. I suggest to show protein content as supplemental material to avoid this criticism.

There are not remarkable changes in protein content between the mutants within each experiment. As stated in the Materials and Methods section samples were normalized by their total protein content and against a control condition in each experiment. We agree with the reviewer and data has been included as supplementary material (Table S3).

7 – Overall, the figure formatting requires improvements. There are Y-axis without line, unnecessary grid lines, number with commas instead of points, etc.).

We appreciate again the critical look at the Figures. We have reformatted the graphs as suggested by the reviewer.

8 – When comparing the genotypes and salt stress (2 factors), I think that a two-way ANOVA would be more adequate than the one-way ANOVA used. It also necessary when comparing the genotypes and carbon source.

The statistical analyses have been repeated taking this suggestion into account and inlcuded in Fig. 1 and Fig. 5A . Two of the comparisons have been kept as Welch’s ANOVA because the data did not meet the two-way ANOVA assumptions of normality or homoscedasticity. Namely, the dataset from Fig.2A violated the homoscedasticity assumption (Levene’s test, W(11, 60)=10.6, P = 1.91x10^-10), while the datasets for Fig.5B and Fig.6E are not normally distributed (Shapiro-Wilk’s test, W = 0.963, P < 0.001; and W = 0.938, P < 0.01, respectively). Welch’s ANOVA is a robust option for non-parametric data, while the two-way ANOVA could introduce type I errors in this case, so we have decided to keep it despite being unable to discern interactions between factors.

9 – Lines 198-199: A statistical analysis of correlation (as linear regression, Pearson’s correlation, etc.) is required to conclude that there is a correlation between the variables. This valid for most of the data of the manuscript to demonstrate that a phenotypic variable (as germination, root lengths, etc.) is indeed correlated to NO level.

Thank you for noticing the need of a statistical analysis of correlation. Indeed, there was a statistically significant negative correlation between germination and NO content. Pearson's correlation coefficient was -0.94 (t10 = -8.97, P = 4.28x10^-06) for t50 and -0.87 (t10 = -5.55, P = 0.0002) for U7525. We have included so in the Results section 2.2.

For the rest of meristem parameters we have performed a PCA analysis that basically shows the same tendency as ANOVAs in terms of correlation, so we would rather prefer not to duplicate this information. (See attached Figure for reviewer).

10 – Line 255: It is not the case of all mutants, as no difference was detected between WT an pOCA108.

pOCA108 is not a mutant, it is the wild-type control line for cue1-1 in the Bensheim (Be-0) ecotype background as described in the Materials and Methods section.

11 – Lines 297-298: It is essential to check whether the higher NO levels of cue1 mutants also occur without the carbon supplementation, otherwise they cannot claim that it is related to the developmental stage (it can be related solely to the sugar treatment). Also, the control without sugar supplementation is required in Figure 6. Or in the other experiments (Figures 1-4), have the medium received carbon supplementation? Please clarify this crucial point.

While it would no doubt be interesting to have information about NO production of cue1 mutants without the carbon supplementation, there is an important methodological constrain. As explained in the results section, in the absence of CUE1, plants are unable to establish photoautotrophic growth if they are not supplemented with exogenous metabolizable sugars, such as sucrose or glucose (See reference 24 and attached Figure for reviewer). As shown in the figure, 7-day-old cue1-1 and nox1 mutant seedlings are severely impaired in growth and development compared to Col-0.

For this important restriction, our experiments were always done with a carbon source supplementation. In other words, the medium that has been used for germination contains 2% glucose and is directly comparable with the experiment of NO quantification and elongation of the primary root at 7 days that also contains 2% glucose. This information is properly provided in the Methods section.

12 – In Figures 5 and 6, I really missed the two mutants that were not analyzed. They should be included here or removed from the whole manuscript, otherwise the discussion and conclusions become confusing and cannot be generalized.

We chose to use only one allele for each background just in Figures 5 and 6 to streamline the experimental design and to make the results more readily understandable. For the Columbia background, appart from the seed germination phenotype, the rest of root related phenotypes are completely comparable and similar between cue1-5 and cue1-6 or nox1, so neither cue1-6 nor nox1 were used in those experiments in order to present a clearer result and to avoid repetitive data.

13 – Also in Figures 5 and 6, the use of cPTIO is necessary to demonstrate that NO level is indeed involved in the observed developmental phenotypes and that they are not related to other modifications that occur in such pleiotropic mutants.

We appreciate this thoughtful comment and suggestion made by the reviewer. Indeed, we have previously published the use of cPTIO in the cue1-1/nox1 backgrounds (please see reference Fernandez-Marcos et al., 2011 PNAS), in terms of root apical meristem defects and root growth inhibition, while cue1-1/nox1 mutation caused decreased PIN-FORMED 1 (PIN1)-dependent acropetal auxin transport.

We have now submitted a new manuscript demonstrating that enhanced NO levels in cue1-1/nox1 mutants are responsible for the observed root developmental phenotypes, since decreased PIN1 expression in cue1 can be rescued by scavenging NO under cPTIO treatments (see attached Figure for reviewer). We would rather prefer to maintain this body of evidences in the already submitted manuscript results.

14 – Lines 371-378: This information has been provided before in the text. It is repetitive.

We appreciate this suggestion. This paragraph has been removed.

15 – Line 383: It is better to use “decrease” rather than “lack” of NO. For me, lack sounds as if there was no NO in the mutants.

We agree with the reviewer and the term decrease has been used.

16 – The mutants have different ecotype backgrounds. However, only Col-0 was used as a WT control. So, how the authors can ensure that the differences between pOCA108-1/cue1-1 and Col-0 are not related to the different ecotypes?

It is a similar concern to the number 10 previously addressed. pOCA108 is not a mutant, it is the wild-type control line for cue1-1 in the Bensheim (Be-0) ecotype background as described in the Materials and Methods section.

17 – NO production was quantified spectrofluorometry using protein extracts, which is not a good procedure. In this condition, tissues and probably organelles are not intact so that many artifacts can occur in DAF fluorescence. A negative control with cPTIO is necessary in this assay (as done in microscopy analysis), particularly in those in Figure 1 (there is no microscopy analysis to compare).

Thank you for the suggestion made by the reviewer. To rule out the possibility that we are detecting artifacts, we have conducted controls with cPTIO in DAF fluorescence during seed germination and early seedling development in our previous published works (Fernández-Marcos et al., PNAS 2011; Sanz et al., Plant Physiology 2014; Albertos et al., Nature Communications 2015), and also in the present manuscript. As it can be seen in the graph of the attached Figure for reviewer file, cPTIO partially eliminated the fluorescence in the pre-loaded samples.

However, there are concerns about the suitability of cPTIO as a fluorescence control because it has been shown that it is only partially successful at scavenging this fluorescence and it can even increase it (Yamasaki et al. 2016/Plant Nitric Oxide: Methods and Protocols, D’Alessandro et al., 2013, Planchet and Kaiser, 2006/JXB), so we decided to include these controls in the manuscript as Supplementary Materials (Figure S1). Given the concerns raised by the reviewer, we decided to undertake the detection of NO through confocal microscopy, using the same experimental conditions and DAF staining protocol that we used for our former spectrofluorometry measurements.

In Figure S1, we are showing the root tip of 7-day-old Columbia-0 WT stained with DAF-FM DA. We are using the FIRE LUT as a fluorescence heatmap. As shown, cPTIO is able to scavenge DAF fluorescence (although not fully suppressed), and GSNO increases local maxima.

This technique is less effective at 4 days after stratification (i.e. seedlings seem to take up the compound less effectively). Please see the attached Figure for reviewer file. However, even with this less effective uptake of DAF-FM DA and the very low sample number, we see similar results to what we showed in the original spectrofluorimetry experiment – a similar level of endogenous NO in all lines and no NO overproduction by the mutants.

Reviewer 3 Report

The submitted manuscript by Lechón et al., presents cue1 as the primary mutant to study the NO signalling pathways in Arabidopsis. The authors demonstrate an association of NO accumulation with several pleiotropic phenotypes including root growth, root architecture, and salinity stress. This was achieved by growing and comparing phenotypes of several cue1 mutants. The authors have further investigated the link between carbon metabolism and NO signaling by growing the mutants with exogenous sugar supplement.

Overall, I must commend the authors on a very well-written and presented manuscript, which was a pleasure to read. The two major comments on the manuscript are:

  1. Figure 1A: Phenotypic differences are not very clear in Figure 1A. The authors might consider using a few representative seedlings for each genotype in a single image with a dark background and a scale bar.
  2. The authors have mentioned that cue1-5 mutant also harbours a mutation in transparent testa/glabrous. cue1-5 mutants also showed the most deviant phenotype among the mutants studied. Why did the authors not test an independent transparent 524 testa/glabrous mutant to differentiate the effects/interference from this mutation on NO signalling or growth-related parameters?

Minor comments

  1. The authors mention in the abstract- “we have sought to improve the characterization of cue1 mutants and establish the experimental conditions under which they overproduce NO.” However, in the manuscript NO content at only 2 seedling stages and salinity stress has been assessed, none of which an increase in NO production. Please edit the abstract to reflect the work conducted in the manuscript.
  2. The title 2.1 “cue1 mutants do not overproduce NO during early post-germinative plant development” is a bit misleading title considering that the mutants overall behaved similarly to controls. Please consider revising of the heading.

Author Response

The submitted manuscript by Lechón et al., presents cue1 as the primary mutant to study the NO signalling pathways in Arabidopsis. The authors demonstrate an association of NO accumulation with several pleiotropic phenotypes including root growth, root architecture, and salinity stress. This was achieved by growing and comparing phenotypes of several cue1 mutants. The authors have further investigated the link between carbon metabolism and NO signaling by growing the mutants with exogenous sugar supplement.

Overall, I must commend the authors on a very well-written and presented manuscript, which was a pleasure to read. The two major comments on the manuscript are:

We thank the reviewer for reserving some of their valuable time to our manuscript. We are grateful for the constructive comments and have adapted the manuscript according to his/her suggestions.

Below, we include a point by point overview to the reviewer's comments.

  1. Figure 1A: Phenotypic differences are not very clear in Figure 1A. The authors might consider using a few representative seedlings for each genotype in a single image with a dark background and a scale bar.We appreciate the critical look at Figure1A. We have included new representative seeds (tt mutation is only visible at this stage but not in seedlings) for each genotype in a dark background and a scale bar.
  2. The authors have mentioned that cue1-5 mutant also harbours a mutation in transparent testa/glabrous. cue1-5 mutants also showed the most deviant phenotype among the mutants studied. Why did the authors not test an independent transparent 524 testa/glabrous mutant to differentiate the effects/interference from this mutation on NO signalling or growth-related parameters?

    We appreciate these critical comments about the cue1-5 additional mutation. We have investigated this possibility but have run into a problem regarding the exact transparent testa mutation of cue1-5. We obtained the cue1-5 seeds from ABRC, and were donated by Prof. Chory, creator of the original mutant. There is no description of this additional mutation either in TAIR or in the original paper (Streatfield et al., 1999/Plant Cell). In Streatfield et al., 1999, it is specified that the cue1-5 mutant corresponds to germplasm CS3156, curated by ABRC. In the corresponding database entry in TAIR the phenotype is described as follows:

    “yellowish plants and reticulated leaves (veins darker than interveinal tissues); yellow seeds (lacks brown pigment in seed coat)”

    The information in TAIR does not distinguish between the reticulated leaf phenotype and the yellow testa, as if both originated from the cue1 mutation. cue1-5 has been published as a weak allele with an Arg to Cys point mutation, but none of the other cue1 alleles have the yellow testa phenotype, so it is unlikely that the yellow testa is caused by this mutation.

    The parent line of this mutant is CS3176, donated by George Redei to the ABRC in 2002. This line comes from George Redei’s line 5-13, created between 1957 and 1965 by an X-ray mutagenesis of a heterogenous Landsberg population isolated in Friedrich Laibach’s lab. This Columbia WT line was mutagenized with EMS between 1965 and 1969, and later commercialized by the company Lehle Seeds, and it is this EMS collection the one used to isolate the cue1-5 mutant.

    According to TAIR, Redei published a paper with cue1-5 back in 1965 (Redei 1965/American Journal of Botany). However, in this paper the only cue1-5 related phenotype that is characterized is reticulata. At the same time, we know that Redei was the first one to identify the transparent testa mutations, although they were not published (Shirley et al., 1995/Plant Journal). In this article, they summarize the known transparent testa mutations known to date, their phenotype and how they were obtained. Of those, the only ones obtained by EMS mutagenesis are tt4 and ttg. Since the progeny of cue1-5 crosses segregates the yellow phenotype in the expected Mendelian ration, we think that if there is indeed an additional mutation it must be close to the locus of CUE1, so we think the mutation must lie in the TTG locus, as it is the closest of those two to CUE1.

    We have previously published the characterization of tt4 mutation (Sanz et al., 2014 Plant Physiology) to differentiate the effects/interference from this mutation on NO signalling and root growth-related parameters, and this is reason to not be inlcuded here.

Minor comments

  1. The authors mention in the abstract- “we have sought to improve the characterization of cue1 mutants and establish the experimental conditions under which they overproduce NO.” However, in the manuscript NO content at only 2 seedling stages and salinity stress has been assessed, none of which an increase in NO production. Please edit the abstract to reflect the work conducted in the manuscript.

    Thank you for noticing that our abstract was not concrete enough. Abstract has been fully modified according to the suggestions of the reviewers.

  2. The title 2.1 “cue1 mutants do not overproduce NO during early post-germinative plant development” is a bit misleading title considering that the mutants overall behaved similarly to controls. Please consider revising of the heading.

This subheading has been corrected to make our statement clearer (cue1 mutants accumulate wild-type NO levels during early post-germinative plant development).

Round 2

Reviewer 2 Report

The authors have answered properly the questions raised by all reviewers, including the revision of statistical analysis and the clarifications about sugar supplementation, wild-type control lines and NO measurement. Despite the limitations and difficulties regarding NO measurement methodology (which is a rather common issue in plant NO research), their data are consistent with the expected phenotypes of the mutants after seedling establishment. 

This manuscript is a resubmission of an earlier submission. The following is a list of the peer review reports and author responses from that submission.

Round 1

Reviewer 1 Report

The purpose of this study is to validate the use of cue1 mutants as genetic NO overproducers.

This is an important attempt because the use of NO donor chemicals may not always replicate the endogenous effects of NO and it is necessary to use genetic tool to evaluate the effect of the endogenously produced NO. The authors characterized cue1 deficient lines. CUE1 encodes a PPT in the plastid membrane, and its deficiency produces not only NO-associated phenotypes but other outcomes. The authors' aim here was to examine whether a distinct phenotype is correlated with the alteration of endogenous NO level in different lines. 

I have two major concerns.

1) In Figure 2 (control experiment), different lines showed different t50 within 60 h after sowing. If the difference in the NO level determines the germination rate, the NO contents of these lines should be different at time points before 60 h, but the provided NO data was for 4-d-old seedling (Figure 1). It is necessary to explain the validity of this experimental design.

2) The description of Abstract is too general. More concrete information should be provided. It is necessary to explain the identity of CUE1 protein, and its relationship with the NO metabolism. Also, names of the mutants used in this work should appear. The conclusion 'cue1 mutants only accumulate NO after germination' is oversimplified and actually invalid because all measurements (including Figure 1) were done only after germination.

Below are minor comments.

3) Figure 2A. Y-axis label should be Gmax, in correspondence with main text.

4) Figure 2 C, D. Font size of axis labels and sample labels is too small.

5) Figure 2 legend (Figures 4, 5 and 6 also). Use capitals A, B... to indicate the figure panels.

6) The quality of Figure 3 is not satisfactory. The actin bands of the both sides (Col-0 in control treatment and cure1-1 in NaCl treatment) are too weak and cannot be taken as control. Ponceau S staining is not necessary. In Coomassie staining image, the range of the molecular size of the displayed area should be indicated. Using 'alpha' for 'anti-' should be avoided because it is confused with alpha-actin.

7) Line 167. 'the stark decrease in NO': It is unclear which difference is mentioned here.

8) Line 214-216. The mechanism described in the later half is a presumption, not the reason. This sentence should be divided into two, to distinguish the fact from the author's idea.

9) Line 266. 'a narrower vascular bundle': Indicate the vascular bundle in the corresponding images in Figure 4B.

10) Line 273. ... a sugar translocator required for proper plastid functions and ...

11) Line 315. 'their root elongation was vastly different': What data is mentioned? Elongation rate, or root length? In which conditions?

12) Line 329. 'the cue1 mutants presented amyloplasts in the columella stem cells': Indicate the columella stem cells in the corresponding images both of wild type and mutants in Figure 6, so that readers can find the difference between the lines clearly.

13) Line 348-349. It is not clear how the authors evaluated the extent of differentiation.

14) Line 352. the RAM size in cue1-5 was 21% smaller ...

15) Table 1 contains the same data presented in Figure 5B and Figure 6C,D. Such overlaps should be avoided.

16) Figure 4 and 6. Provide more close-up root tip figures, to allow readers to examine the cell shape and size. Put scale bars for each magnification. Font size of the labels is too small.

Reviewer 2 Report

The manuscript attempted to reveal the relationship between nitric oxide (NO) accumulation and phenotypical changes at early developmental stages, by utilizing 4 different cue1 mutants, which had been demonstrated to overproduce endogenous NO previously. Here, the endogenous NO content was monitored by a pharmacological strategy, during seed germination and root elongation stage, as well as salt stress conditions. Lastly, the authors concluded that cue1 mutant could be considered as a useful tool to study physiological functions of NO, at certain developmental conditions.

Overall, the current study was designed and performed correctly. The results are clear and exhaustive. However, in my opinion, they study did not provide sufficient data to support their main conclusions. Below, I outline a few concerns that the authors might want to address in a revised manuscript:

1.      To date, NO biosynthesis and signaling have been investigated extensively. Several recent reviews (e.g. Hasanuzzaman et al. 2018/Plant Biotechnology Report; Begara-Morales et al. 2018/JXB) provide constructive insights to decipher the function of NO as an essential biological messenger involved in plant response to stress. It is very helpful to provide readers a brief summary of the most recent findings of NO.

2.      The Figures as presented in the manuscript appear to be of low quality. The font size is too small making labels difficult to read. In “result 2.2”, it is indicated that “the maximum germination of cue1-5 was only 67%” at day 4 after sowing (also displayed in Figure 2A). Whereas in Figure 2C, according to the “cumulative germination slope”, the germination rate of cue1-5 at 96 hours (=4 days) after sowing is approximate 90%.  Could the author explain the reason?

3.      The author pinpointed that in the cue1-5 mutant contained an additional insertion at TTG1 gene, which significantly influence carbon partitioning in Arabidopsis seeds (Li et al. 2018/Nature communication). It would be interesting/important to compare the phenotype (as well as NO content) between cue1-5 and ttg1 mutant, in order to rule out the background interference.

4.      In “result 2.5”, sugar (2% glucose or 0.75% sucrose) was supplemented in the MS medium, in order to examine the relationship between carbon metabolism and NO homeostasis. To reveal the facts, I think it is necessary to design the experiment by supplementing the MS medium with a sugar gradient (titration). Without such a gradient, it is hasty to conclude with current data.

5.      In “result 2.5”, I am wondering why the nox1 mutant is excluded from the experiment.

6.      Each panel in figure 3 should be well organized and labelled. It appears that the loading amount of protein is not equivalent between “Control” and “NaCl”, based on the “Coomassie B250” staining results. If indeed the loading is not equivalent, it should be clearly indicated in Figure Legend.

7.      In figure 4B and 6A&B, a scale bar should be included in each picture.

8.      In figure 5B, how many samples/roots have been examined? (n=?)

9.      In Table 1, “Average Cell Length” lacks units. (µm?) Also, for the “n=104”, does n = 104 roots or the number of cells measured?

10.  In figure1, 2 and 3, “MS ± 100 mM NaCl” should be changed to “with and without 100 mM NaCl”.

11.  In the main text and figure legend, 2% Glucose or 0.75% sucrose should indicated as 2% glucose (w/v).

12.  In the figure 6C, D & E, were the data extracted from Table 1, or were they generated from another set of experiments.

13.   In “Materials and Methods”, “4.4 western blotting”, how many tissues were used for grinding?

Reviewer 3 Report

The manuscript of Lechon et al. addresses NO production and homeostasis during plant development and stress responses. Authors have used various Arabidopsis cue mutants, which have been previously associated with NO overproduction and/or NO accumulation. Authors analysed phenotypes of four different cue1 alleles during seed germination, primary root elongation and responses to salt stress. This is an area of potential high interest to a wide audience of researchers involved in plant NO research; however, on my opinion, the major part of results and their discussion are in a large extent flawed by methodological deficiencies of NO analysis.

Authors aimed to study NO levels and homeostasis by determining “NO content” in plant extracts by a spectrofluorimetric method using a commercial probe DAF-FM DA.  First of all, it should be noted that although in their Introduction part of the manuscript authors refer to multiple methodological problems associated with the pharmacological approach using so-called “NO donors”, they neglect similar deficiencies described in the literature for the use of so-called “specific NO probes”. Among these, the wide use of diaminofluorescein (DAF) probes, where DAF-FM belongs, has been criticized as non-specific and suffering from diverse interferences both in animal and plant samples - for review on NO methods see e.g. Csonka et al. British Journal of Pharmacology (2015) 172 1620–1632. Thus, a great care must be taken in the methodology (proper use of control samples etc.) and data interpretation and discussion – clearly, this has not been done in the present study. Secondly, cell-permeable diacetyl derivatives were design to enable in vivo NO imaging (which is far from being quantitative, as mentioned above). It makes no sense to use cell-permeable variant, DA-FM DA, to try to measure NO in extracted samples, and then call it, as authors did, “endogenous NO”. Authors here refer to the paper of  Pérez-Chaca et al. (ref.38), but effectively, in the cited study, DAF-2 DA probe (a cheaper but less reliable compound compared to DAF-FM DA) was used for the microscopical analysis whereas DAF-2 probe for the spectrofluorimetric analysis. Finally, authors describe their methods that “Samples were incubated at 37°C for 2 hours in the dark.”. NO is highly reactive compounds and multiple enzyme and non-enzyme reactions occurring in a buffered aqueous extract, consuming or producing NO, can occur in an irreproducible manner within 2 hours. So, even if the used method would be specific and accurate (which is not), it would give erroneous results in respect to “endogenous NO content” in original plant samples before the extraction procedure.

As a last but relevant point, it is worth noting that the same method was used in the original report on nox1/cue mutants (He et al. Science 2004, 305, 1968–71) and it should be noted that the same senior author, involved in this study, Nigel Crawford, was involved as senior author in largely disputed papers on AtNOS (honestly, which should had been retracted, on my opinion) by Guo et al. (F.-Q. Guo, M. Okamoto, N. M. Crawford, Science 302, 100 (2003).

In conclusion, if authors aim to measure “endogenous NO content” in the context of their experiments, they need to use some appropriate method currently available, ie. EPR, NO electrodes or in-vivo NO imaging.

I would be also very sceptical to authors results and conclusions on the quantification of protein levels of ABI5 by Western blot method. It is strange that authors used 4% SDS-acrylamide/bisacrylamide gels, which is usually the gel density used in stacking gel, whereas 7-12% density is usual in separation PAGE gels. Moreover, there are evidently changing amounts of actin, used as a loading control, which furthermore shows a different trend compared to Ponceau and CBB staining. So on my opinion, any quantitative conclusions from presented blots are impossible.

As a minor comment, it is not correct to state that “… a lot of progress has been made since it was first discovered that NO was produced by plants, twenty years ago” (Abstract, L10-11) or that “Since the establishment of nitric oxide (NO) as an endogenous signaling molecule in plants twenty years ago …” (Introduction, L28-29). The first report on NO production in plants by Klepper is dated in 1979, ie. 40 years ago (Klepper LA. 1979. Nitric oxide (NO) and nitrogen dioxide emissions from herbicide-treated soybean plants. Atmospheric Environment 13, 537-542). Other publication reported on NO role in various plant processes, including plant pathogenesis and symbiosis (see eg. Laxalt AM, Beligni MV, Lamattina L (1997) Nitric oxide preserves the level of chlorophyll in potato leaves infected by Phytophthora infestans. Eur J Plant Pathol 103: 643 or Cueto M, Hernández-Perera O, Martín R, Bentura ML, Rodrigo J, Lamas S, Golvano MP (1996) Presence of nitric oxide synthase activity in roots and nodules of Lupinus albus. FEBS Lett. 398:159-64) prior to the seminal papers of NO role in pathogenesis, cited as reference 2 and 3 of the present manuscript. Authors shall rewrite these sentences to reflect accurately the historical background of plant NO field (the paper by Dangl cited as ref.1 is just a follow–up to these two original articles).

Reviewer 4 Report

Manuscript ID.: plants-485918
Title: Differentail nitric oxide (NO) overproduction by cue1 mutants during early plant development

The authors assessed phenotype of four different cue1 alleles, which are mutants overexpression NO, during physiological processes such as germination, primary root elongation and salt stress tolerance. This is an interesting subject with potentially important findings. However, even well-targeted measurements, I have very little confidence in the scientific significance (e.g., novel finding, impacts to other studies etc.), In addition, the authors didn’t provide clear evidence why cue1 mutants present low levels of NO, how cue1 mutants inhibit root development despite of low levels of NO, and whether carobhydate-mediated NO is related to other mechanism during early plant development, even though it was explained with many references.

Therefore, the present manuscript is not acceptable for publication in Plants, encouraging substantial revision. 

General comments
1. The title does not precisely reflect the findings of the manuscript. Authors may consider, if possible, to add a statement of the main result or conclusion presented in the manuscript.
2. The objectives of this work were established the conditions in which the cue1 mutants overproduce NO. What does it mean by the “condition”? The objectives of this study should clearly be noticed.
3. During seed imbibition, cue1 mutant showed low NO levels despite of NO overproducing mutants. The interaction between flavonoid synthesis and oxygen during seed imbibition should be more clearly addressed. In this context, author should check the flavonoid and ROS in seed itself and their effects on NO level during seed imbibition, even though the authors tried to explain this interaction with previous studies which were carried out after seed germination and seedling stage.

4. NO has been generally known to promote seed germination through inhibition of ABA accumulation by binding ABA catabolism protein as like CYP707A2. The present results show that cue1-5 mutant having the lowest NO level promotes ABI5 gene expression which is downstream of ABA, consequently inhibition of seed germination. This phenomenon was prompted under salt stress condition, as shown more decrease of NO and germination. But ABI5 expression was strongly appeared in all of salt stressed seed. These are unclear between NO content and ABA-mediated inhibition of germination.

5. NO is closely related to inhibition of root elongation in primary roots but not in lateral roots during seed germination. In the present manuscript, however, root elongation was not improved in cue1 mutants showing low NO content during seed germination. Author suggest this is lack of sugar source in cue1 mutants due to defective in chloroplast biogenesis. The authors thus applied sucrose and glucose as carbon source, and found low NO content and root elongation in cue1 mutants. They suggest NO accumulation depends on sugar metabolism. However, the following questions should be answered: What are the signalling factors of root elongation at seed imbibition (which showed low NO without carbon source)? Are there any other mechanisms associated with carbohydrate-mediated NO?

Minor comments
Abstract
1. Main findings (the interaction between NO and germination or root elongation or ABAI5 or carbon metabolism, in particular) should be more presented, diminishing the background description.
Methods
1. To evaluate the interaction NO and sugars, the authors used the concentration of 2% glucose or 0.75% sucrose for treatments? However, I am not found your experiment pre-test the concentration of sugars or based on the previously study. Please add the references?

Results
1. Describe the results in concise manner.
2. In Figure 3, actin was used loading control for ABI5 accumulation by Western blot but it was not same between control and the different types of NO mutants. Please, check it again?
Discussion
1. The Discussion is not adequate and some parts were not matched well with the results obtained. This section should be fine trimming accordingly the results with proper interpretation.
2. Why the NO level was increased in the cue1-5 line in the presence of glucose and sucrose?  Authors should address the phenomena in detail.